# Antimicrobial Activity of Gemini Surfactants with Ether Group in the Spacer Part

**DOI:** 10.3390/molecules26195759

**Published:** 2021-09-23

**Authors:** Bogumil Eugene Brycki, Adrianna Szulc, Iwona Kowalczyk, Anna Koziróg, Ewelina Sobolewska

**Affiliations:** 1Department of Bioactive Products, Faculty of Chemistry, Adam Mickiewicz University Poznan, 61-614 Poznan, Poland; adaszulc@amu.edu.pl (A.S.); iwkow@amu.edu.pl (I.K.); 2Institute of Fermentation Technology and Microbiology, Faculty of Biotechnology and Food Science, Lodz University of Technology, 90-924 Lodz, Poland; anna.kozirog@p.lodz.pl; 3Interdisciplinary Doctoral School of the Lodz University of Technology, Lodz University of Technology, 90-924 Lodz, Poland; ewelina.sobolewska@dokt.p.lodz.pl

**Keywords:** gemini surfactants, functionalized spacer, antimicrobial activity, MIC, surface activity

## Abstract

Due to their large possibility of the structure modification, alkylammonium gemini surfactants are a rapidly growing class of compounds. They exhibit significant surface, aggregation and antimicrobial properties. Due to the fact that, in order to achieve the desired utility effect, the minimal concentration of compounds are used, they are in line with the principle of greenolution (green evolution) in chemistry. In this study, we present innovative synthesis of the homologous series of gemini surfactants modified at the spacer by the ether group, i.e., 3-oxa-1,5-pentane-*bis*(*N*-alkyl-*N*,*N*-dimethylammonium bromides). The critical micelle concentrations were determined. The minimal inhibitory concentrations of the synthesized compounds were determined against bacteria *Escherichia coli* ATCC 10536 and *Staphylococcus aureus* ATCC 6538; yeast *Candida albicans* ATCC 10231; and molds *Aspergillus niger* ATCC 16401 and *Penicillium chrysogenum* ATCC 60739. We also investigated the relationship between antimicrobial activity and alkyl chain length or the nature of the spacer. The obtained results indicate that the synthesized compounds are effective microbicides with a broad spectrum of biocidal activity.

## 1. Introduction

Antimicrobials are compounds that, apart from antibiotics, are used to reduce the population of microorganisms or infectious agents to a non-threatening level [1,2]. These actions are extremely important, especially at the time of the COVID-19 pandemic (coronavirus disease of 2019) caused by SARS-CoV-2 (severe acute respiratory syndrome coronavirus 2), which increased people’s awareness of pathogens, the need for effective use of microbiocides, and the demand for these compounds [3]. It was reported that the global biocides market size is expected to reach 10.5 billion USD by 2027 [3]. Nowadays, it is not only COVID-19 a serious epidemiological threat but also the formation of resistant bacteria strains, which intensifies in the face of climatic changes [4,5]. For this reason, research of a new generation of antimicrobials as effective agents in the fight against microorganisms has become imperative [6]. Microbiocides, in terms of chemicals, are classified as oxidizing substances, alcohols, phenols and their derivatives, aldehydes, organic and inorganic acids, halogen compounds, and quaternary ammonium compounds [7,8,9]. They are used as antiseptics, disinfectants, and preservatives in a wide variety of fields: water purification, personal care products, food and beverage, wood preservation, paints and coatings, plastics, cleaning products, ventilation and air conditioning, medical products, and also in oil, gas, fuel, and paper industry [3,10].

The most important and rapid developing group of antimicrobials includes the quaternary ammonium compounds (QAC) [11]. Their effectiveness has been proven over the past century, and various new generations have been introduced since then [11,12,13]. In the second half of the twentieth century, double quaternary ammonium salts were obtained, and named gemini surfactants (GS) by Menger in 1991 [14]. Since then, this class of compounds has been very intensively developed. Gemini surfactants are constructed from two long hydrocarbon substituents and two cationic, ammonium head-groups linked by a spacer. The linker may be long or short, flexible or rigid, and polar or non-polar, but most often it is a chain of methylene groups, sometimes functionalized with groups containing oxygen or nitrogen atoms or with aromatic rings [1,15,16,17,18,19,20]. The possibility of modifying the structure within the linker, substituents, counter ion, or the cation environment makes it possible to design the structure of the compound so that it exhibits the desired utility properties [1]. The presence of two cations in the molecule causes gemini surfactants to possess several times better properties than their monomeric counterparts [1,21,22,23,24,25]. They are also known for their broad spectrum of action against bacteria [26,27,28,29,30,31], fungi [32,33,34,35,36,37], and algae [38]. QAC are significant antiviral compounds [39,40,41,42,43,44]; therefore, it can be assumed that the effectiveness of gemini surfactants against viruses will also be high. The unique antimicrobial effectiveness of gemini surfactants can be explained by the mechanism of biocidal action, which consists in adsorption of cations on the surface of the microorganism and penetration by long alkyl substituents of the cell membrane. This results in damage and outflow of the cytoplasm along with potassium ions and other cellular components, resulting in the death of the microbial cell [28,45,46,47]. Gemini surfactants are named as multifunctional because they also show high aggregation, surface, and anti-corrosive activity [1,25,48,49,50,51,52]. Moreover, compared to QAC, they are less toxic with respect to aquatic organisms [53,54]. Gemini surfactants, due to their excellent antimicrobial activity, are hardly biodegradable, but there are solutions to improve biodegradability, such as the introduction of labile bonds [55,56,57,58,59,60] or the use of immobilized consortium of microorganisms in Ca-alginate beads [61,62]. Gemini surfactants have one more unique features that makes them superior to QAC; they are not susceptible to current QAC bacterial resistance machinery. Due to conformational flexibility, these compounds tend to escape such resistance mechanisms [27].

In reference to the need to discover new microbicides with a broad spectrum of action, we focused on the synthesis of gemini surfactants functionalized with ether group in spacer part. We have also paid particular attention to the determination of relationship between antimicrobial activity against bacteria or fungi and the structure of GS. In this study, detailed spectroscopic analysis and the surface activity of obtained gemini surfactants were also investigated.

## 2. Results and Discussion

### 2.1. Synthesis and Analysis

3-oxa-1,5-pentane-*bis*(*N*-alkyl-*N*,*N*-dimethylammonium bromides) were obtained by innovative synthesis without a solvent (Figure 1), according to the preparation developed in our laboratory, described previously [63]. This method of obtaining gemini surfactants ensures the best results in the shortest period of time. The synthesis proceeds according to the mechanism of nucleophilic substitution S_N_2, which proceeds most quickly when the concentrations of the reagents are the highest, and they are at the highest under solvent-free conditions. In addition, this type of preparation ensures the highest purity of the obtained product and the absence of by-products. This is in good correlation with the greenolution idea in chemistry [64,65].

The purities for all products were confirmed by elemental analysis (EA):

*3-oxa-1,5-pentane-bis(N-butyl-N,N-dimethylammonium bromide)* (**4-*O*-4**), m.p. 224–225 °C; EA (%) for C_16_H_38_Br_2_N_2_O: found C 44.42, H 8.88, N 6.30, calcd. C 44.25, H 8.82, N 6.45;

*3-oxa-1,5-pentane-bis(N-hexyl-N,N-dimethylammonium bromide)* (**6-*O-*6**),m.p. 220–222 °C; EA (%) for C_20_H_46_Br_2_N_2_O: found C 48.82, H 9.50, N 5.50, calcd. C 48.98, H 9.45, N 5.71;

*3-oxa-1,5-pentane-bis(N-octyl-N,N-dimethylammonium bromide)* (**8-*O*-8**), m.p. 242–243 °C; EA (%) for C_24_H_54_Br_2_N_2_O: found C 51.96, H 9.91, N 4.73, calcd. C 52.75, H 9.96, N 5.13;

*3-oxa-1,5-pentane-bis(N-decyl-N,N-dimethylammonium bromide)* (**10-*O*-10**), m.p. 249–250 °C; EA (%) for C_28_H_62_Br_2_N_2_O: found C 56.49, H 10.62, N 4.33, calcd. C 55.81, H 10.37, N 4.65;

*3-oxa-1,5-pentane-bis(N-dodecyl-N,N-dimethylammonium bromide)* (**12-*O*-12**), m.p. 248–249 °C; EA (%) for C_32_H_70_Br_2_N_2_O: found C 58.48, H 11.19, N 4.15, calcd. C 58.35, H 10.71, N 4.25;

*3-oxa-1,5-pentane-bis(N-tetradecyl-N,N-dimethylammonium bromide)* (**14-*O*-14**), m.p. 246–247 °C; EA (%) for C_36_H_78_Br_2_N_2_O: found C 60.48, H 11.41, N 3.70, calcd. C 60.49, H 11.00, N 3.92;

*3-oxa-1,5-pentane-bis(N-hexadecyl-N,N-dimethylammonium bromide)* (**16-*O-*16**), m.p. 240–242 °C; EA (%) for C_40_H_86_Br_2_N_2_O: found C 61.41, H 11.68, N 3.20, calcd. C 62.32, H 11.24, N 3.63;

*3-oxa-1,5-pentane-bis(N-octadecyl-N,N-dimethylammonium bromide)* (**18-*O*-18**), m.p. 236–239 °C; EA (%) for C_44_H_94_Br_2_N_2_O: found C 64.68, H 12.14, N 3.55, calcd. C 63.90, H 11.46, N 3.39.

The purities of synthesized gemini surfactants were also confirmed by spectroscopy methods (^1^H NMR, ^13^C NMR, FTIR). Proton and carbon chemical shifts are listed in Table 1 and Table 2, respectively. NMR and FTIR spectra are given in Appendix A.

The Fourier transform infrared spectroscopy (FTIR) spectra for 3-oxa-1,5-pentane-*bis*(*N*-alkyl-*N*,*N*-dimethylammonium bromides) have typical bands for the stretching asymmetric and symmetric vibrations of CH_3_ and CH_2_ groups, as well as bands for the deformation vibration of these groups. In the correct regions, there are observed bands for the vibration of C-O bonds and skeletal vibrations γ. Increasing the length of the alkyl substituent does not affect the shape of the FTIR spectrum:

**4-*O*-4** FTIR (ν_max_, cm^−1^): 2963, 2874, 1475, 1360, 1120, 1077, 980, 715;

**6-*O*-6** FTIR (ν_max_, cm^−1^): 2980, 2800, 1470, 1364, 1127, 1065, 965, 727;

**8-*O*-8** FTIR (ν_max_, cm^−1^): 2929, 2849, 1466, 1361, 1132, 1066, 930, 713;

**10-*O*-10** FTIR (ν_max_, cm^−1^): 2980, 2800, 1470, 1364, 1154, 1077, 955, 727;

**12-*O*-12** FTIR (ν_max_, cm^−1^): 2913, 2851, 1472, 1351, 1137, 1068, 928, 714;

**14-*O*-14** FTIR (ν_max_, cm^−1^): 2919, 2847, 1468, 1352, 1152, 1050, 978, 713;

**16-*O*-16** FTIR (ν_max_, cm^−1^): 2980, 2800, 1470, 1356, 1127, 1055, 968, 716;

**18-*O*-18** FTIR (ν_max_, cm^−1^): 2951, 2851, 1469, 1359, 1145, 1063, 979, 720.

### 2.2. Surface Properties of Gemini Surfactants

The primary ability of GS is surface activity. This kind of compounds are very effective in lowering surface and interfacial tensions. Surface tension of pure water is approximately 72 mN·m^−1^ [66], and ordinary QAC—DTAB (*N*-dodecyl-*N*,*N*,*N*-trimethylammonium bromide) lowers this value to 36.4 mN·m^−1^, while the dimeric counterpart—12-2-12 (ethylene-1,2-*bis*(*N*-dodecyl-*N*,*N*-dimethylammonium bromide)—is able to achieve 30.64 mN·m^−1^ [67]. It is related to the structure of gemini surfactants and mechanism of surface action in which molecules of surfactant adsorb so that the hydrophilic parts of the molecules are directed towards the polar phase, and hydrophobic groups are directed in a nonpolar phase [1]. The aggregation capacity of QAC is mostly controlled by the intermolecular interaction of surfactants molecules and with solvents. In the case of gemini surfactants, these behaviors are controlled by the cooperativity of intermolecular and intramolecular interactions of GS in addition to their interactions with solvents [68]. Surfactant molecules dispersed in a liquid can form aggregates called micelles. It is characteristic for GS that these aggregates can have different shapes depending on the chemical structure of compound, which determines their properties [16,69,70,71,72]. Gemini surfactants exhibit excellent wetting, foaming, solubilizing, dispersing, and emulsifying properties, and are better than the corresponding QAC every time [73,74,75,76].

The keystone of all surfactant research is delimitation of their CMC (critical micelle concentration), i.e., the smallest concentration at which surfactants molecules rapidly aggregate into micelles [16]. In any case, the CMC values of gemini surfactants are lower than that of the analogue QAC [50,67,77,78,79,80], e.g., the CMC for 16-2-16 (ethylene-1,2-*bis*(*N*-hexadecyl-*N*,*N*-dimethylammonium bromide) is 0.003 mM, while it is 1 mM for monomeric analogue (*N*-hexadecyl-*N*,*N*,*N*-trimethylammonium bromide) [73]. 

The values of CMC, degree of counterion binding (β), and standard Gibbs energy of micellization (ΔG^0^_mic_) for all 3-oxa-1,5-pentane-*bis*(*N*-alkyl-*N*,*N*-dimethylammonium bromides) are presented in Table 3.

There are several factors influencing the value of CMC of cationic GS, such as the following: temperature, chemical structure, and the presence of other substances in the solution [1]. Construction elements affecting CMC values are the length or nature of the spacer [68,81,82,83,84,85,86,87] and the kind of counter ion [69,88,89]. However, the length of hydrocarbon substituent affects the greatest impact on surface activity [90,91,92,93,94,95,96,97,98]. Generally, CMC decreases as the hydrophobicity of the substituent increases. Initially, with the elongation of chain length, the activity dramatically increases. The growth of two methylene groups in the substituent part causes a decrease in CMC values by one order of magnitude. The compounds with very long substituents (hexadecyl and octadecyl) have similar CMC values and aggregation activity. The conclusions presented by us are analogous to previously described results for gemini surfactants [68,80,95,99]. 

The counter ion binding parameter (β) for obtained GS increased with the elongation of substituents up to the compound with dodecyl chain and then decreases. The high the value of β, the stronger the binding of counter ions to micelles and the higher the surface charge density will be [90]. Negative standard Gibbs energy suggests the spontaneity of the micellization process [91,100]. For all of the synthesized GS, ΔG^0^_mic_ takes a negative sign and lower values with the increase in hydrocarbon chains, indicating that the aggregation process is driven by the hydrophobic parts of surfactants. This correlation is characteristic for gemini surfactants [68,101].

It is worth emphasizing that the calculation of critical micelle concentration is also important, due to ecological and performance properties. Toxicity of GS increased when their concentration surpassed CMC [102,103]. Devinsky et al. stated that, for the homologous series of dimeric alkylammonium salts, the best antimicrobial properties possess compounds with average values of CMC [104,105]. 

### 2.3. Antimicrobial Activity of Gemini Surfactants

A very useful and most commonly used parameter for antimicrobial activity of the microbiocides comparison is minimum inhibitory concentration (MIC), i.e., the lowest concentration of compound inhibiting the visible growth of microorganisms after incubation [1,106,107]. MIC values depend on several factors: concentration of active agent, time of the contact, pH, temperature, the presence of organic matter or other compounds, as well as nature, numbers, location, and condition of the microorganism [1]. The main factor determining the antimicrobial activity of gemini surfactants is chemical structure. There are many research studies describing the change of MIC depending on chain length [27,104,108,109,110], functionalization of substituent [75,111], number of methylene groups in spacer [26,67,112,113,114], nature of spacer [115,116,117,118,119], and kind of counter ion [31,120]. The antimicrobial activities of obtained gemini surfactants were determined and listed in Table 4. 

From the obtained results, it can be stated that GS with ether groups in the spacer are more active against Gram (+) bacteria than Gram (−) bacteria. This is in line with the general relationship for gemini surfactants [26,67,121]. Dimeric alkylammonium salts belong to membrane active microbiocides. Due to the fact that the outer cell wall of microorganisms is negatively charged, gemini surfactants can effectively interact with it and adsorb on it. The first effect of microbial interaction is change in cell hydrophobicity, which changes the molecular organization in the membrane. GS are thought to damage the cell wall and outer membrane, resulting in the outflow of low-component cytoplasmic elements and the death of the microorganism [112,122,123]. Gram (+) bacteria possess a simple cell wall composed of a peptidoglycan layer, which permits penetration. The external layer of the outer membrane of Gram (−) bacteria is made up of lipopolysaccharides and proteins, which render the entrance of microbiocides difficult [115]. This explains, in almost every case, the two times higher MIC values against Gram-negative rots *Eschcerichia coli* than the Gram-positive bacteria *Staphylococcus aureus.* Next, yeast with a representative of *Candida albicans* exist in a range of sensitivities. The least sensitive microorganisms to action of 3-oxa-1,5-pentane-*bis*(*N*-alkyl-*N*,*N*-dimethylammonium bromides) are molds, especially *Aspergillus niger*. The MIC value for this strain for the most active compound 12-*O*-12 is 0.117 mM, and it is over 30 times higher compared to the action of the same compound on bacteria. Fungi belonging to *Aspergillus* genus are known for their resistance to chemical disinfectants, which make them very difficult to remove from the surface and air [114].

Compounds with the shortest alkyl chain (4-*O*-4) show the highest values of MIC, and minimal antimicrobial activity. The butyl substituent is too short to effectively penetrate the outer shell of the microorganism’s cells. An increase in hydrocarbon substituent results in an increase in antimicrobial activity (lowering of MIC value). Compound 12-*O*-12 proved to be the most active against all tested microorganisms. Further increase in the hydrophobic chain results in a decrease in activity (increasing of MIC value). Compounds with the longest substituent (18-*O*-18) have over 125, 250 and 4 timer higher MIC than 12-*O*-12 against bacteria, molds, and yeast, respectively. Antimicrobial efficiencies of homologues series of long chain surfactants show a non-linear dependence on chain length. This dependence is quasi parabolic [124]. Devinsky, who is the undisputed pioneer of research into the biocidal activity of double ammonium salts, named this cut-off effect [124,125]. The increase in the lipophilicity of the molecule upon elongating the chain from 12 to 18 carbon atoms causes limited solubility in the aqueous phase and makes those compounds less active. For the obtained GS, cut-off effects can be shown for antibacterial (Figure 2) and antifungal (Figure 3) activity. In addition, in Figure 2, the relationships between MIC and the number of carbon atoms in substituent for homogeneous series of the obtained products and for conventional gemini surfactants bearing ethylene spacers (*n*-2-*n*; 1,2-ethylene-*bis*(*N*-alkyl-*N*,*N*-dimethylammonium bromides) [27] were presented. It can be concluded that the lowest MIC values are observed for compounds containing 10 to 16 carbon atoms in the alkyl chain, regardless of the construction of the linker. Introduction of an ether group into spacer does not reduce the antimicrobial activity of the GS. 3-oxa-1,5-pentane-*bis*(*N*-alkyl-*N*,*N*-dimethylammonium bromides) bearing 10 to 16 carbon atoms in the substituents act as promising antibacterial agents, and their activity can be compared to the activity of classical GS. 

Modification in spacer part of GS consists not only in the introduction of groups imparting hydrophilicity but it is also possible to introduce an aromatic ring, which stiffens the system [1]. In Figure 4, the structural formulas of conventional GS (12-6-12) and compounds with modification in spacers are presented. The antimicrobial activity of these compounds are listed in Table 5.

By comparing the antimicrobial activity of the synthetized gemini surfactants with analogous compounds with a different spacer structure, it can be concluded that the obtained results are consistent, and the obtained MIC values are of the same order. Compounds bearing functionalized spacers by ether or amine group show very similar antibacterial and antifungal activity. Very close MIC values were also received for 12-6-12. Compounds 12-*O*-12, 12-*N*-12 and 12-6-12 have a similar molecular surface. Therefore, it can be concluded that the introduction of an electron-rich group into the spacer does not result in better adsorption of the microbiocide on the surface of the microorganism’s cell. Therefore, it does not involve an increase in antimicrobial activity. On the other hand, obtained GS possess better antibacterial activity than compared to analogic compounds with a stiff spacer constructed from an aromatic ring. Compound 12-Ph-12 shows more than three times higher MIC values against *S. aureus* and *E. coli* than compared to 12-*O*-12. 

It is worth emphasizing that the obtained GS show much greater antimicrobial activity than their monomeric counterparts. The MIC Values of DTAB are 0.252, 0.36, and 0.5 mM against *S. aureus*, *E. coli,* and *C. albicans,* respectively [22,112]. 12-*O*-12 proved to be at least 30 times more active than DTAB. 

## 3. Materials and Methods

### 3.1. Materials 

*N*-butyl-*N*,*N*-dimethylamine (99%), *N*-hexyl-*N*,*N*-dimethylamine (98%), *N*-octyl-*N*,*N*-dimethylamine (95%), *N*-decyl-*N*,*N*-dimethylamine (≥90%), *N*-dodecyl*-N*,*N*-dimethylamine (97%), *N*-tetradecyl-*N*,*N*-dimethylamine (≥95%), and *N*-hexadecyl-*N*,*N*-dimethylamine (≥95%) were obtained from Sigma-Aldrich (Poznan, Poland). *Bis*(2-bromoethyl)ether (≥99%) was purchased from Sage Chemicals (Hangzhou, China). *N*-octadecyl-*N*,*N*-dimethylamine (>90%) was obtained from TCI (Tokyo, Japan). Acetonitrile (≥99%), methanol (≥99%), and phosphorus pentoxide (98%) were purchased from VWR Chemicals (Gdansk, Poland).

### 3.2. Synthesis

In every case, 1 equivalent of *bis*(2-bromoethyl)ether was mixed with 2 equivalents of suitable tertiary amine (*N*-butyl-*N*,*N*-dimethylamine for 4-*O*-4; *N*-hexyl-*N*,*N*-dimethylamine for 6-*O*-6; *N*-octyl-*N*,*N*-dimethylamine for 8-*O*-8; *N*-decyl-*N*,*N*-dimethylamine for 10-*O*-10; *N*-docecyl-*N*,*N*-dimethylamine for 12-*O*-12; *N*-tetradecyl-*N*,*N*-dimethylamine for 14-*O*-14; *N*-hexadecyl-*N*,*N*-dimethylamine for 16-*O*-16; and *N*-octadecyl-*N*,*N*-dimethylamine for 18-*O*-18). The synthetic details are provided in the Appendix A. The reactions were carried out without a solvent, at room temperature, by stirring using a magnetic stirrer until the reaction mixture solidified (approximately 2 h). The exception was the synthesis of 18-*O*-18 due to melting point of amine. In this case, the reaction mixture was heated to 30 °C in a water bath. The reaction was continued until it solidified (4 h). The crude products were crystallized from a mixture of acetonitrile:methanol in the volume ratio of 10:1 and dried in an incubator (60 °C) and over P_4_O_10_ in a vacuum desiccator.

### 3.3. Analytical Methods

The melting point (MP) of the products was measured on Stuart SMP30 (Staffordshire, UK) by using a one-sealed side capillary. 

The elemental analysis (EA) measurements were carried out on a FLASH 2000 elemental analyzer (Delft, The Netherlands) with a thermal conductivity detector.

The NMR spectra for the synthesized compounds were obtained by using a Varian model VNMR-S 400 MHz (Oxford, UK) operating at 403 and 101 MHz for ^1^H and ^13^C, respectively, by using the software, VNMR VERSION 2.3 REVISION A (Varian, Oxford, UK). The ^13^C and ^1^H chemical shifts were measured in CDCl_3_ with TMS as an internal standard. 

The FTIR spectra for the synthesized gemini surfactants were performed using a FT-IR Bruker IFS 66v/S (Poznan, Poland) apparatus. All of the samples were tested in a solid state in the form of tablets with potassium bromide.

### 3.4. Conductivity Measurement

Critical micelle concentration (CMC) values were obtained conductometrically by using a Conductivty Meter Elmetron CC-505 (Zabrze, Poland). The apparatus was calibrated by using a standard (147 µS/cm in 298.15 K). All the solutions were prepared using double-distilled water. Conductivity measurements were carried out at a temperature of 298.15 K. The conductometric titration was repeated at least three times for each gemini surfactant, and CMC was calculated as the mean value of three measurements.

The CMCs of synthesised 3-oxa-1,5-pentane-*bis*(*N*-alkyl-*N*,*N*-dimethylammonium bromides) were obtained by conductometric titration, creating relationship graphs of the characteristic conductivity in water of the GS as a function of the concentration [1,63,69,126]. The graphs consist of two lines with differing slopes. The line with higher inclination shows behavior before micellization, and the second line illustrates the process of micellization. The CMC values are at the intersection of the linear regressions of these lines. The degree of counterion binding (β) was calculated according to Frahm’s method [80] as (1 − α), where α = S_micellar_/S_premicellar_, i.e., the ratio of the slope after and before CMC. The ΔG^0^_mic_ values were calculated by using Equation (1) [48,127]:(1)ΔG0mic=2RT(12+β)lnCMC−RTln2
where R is the gas constant, T is the temperature in K, and the CMC is in mol/L.

### 3.5. Antimicrobial Activity

The gemini compounds were tested for antimicrobial activity against bacteria: *Escherichia* coli ATCC 10536, *Staphylococcus aureus* ATCC 6538, yeast *Candida albicans* ATCC 10231, and molds *Aspergillus niger* ATCC 16401 and *Penicillium chrysogenum* ATCC 60739. The MIC values for all microorganisms were determined by a tube standard two-fold dilution method [114]. Each of microorganisms was resuspended in physiological salt solution (molds in water with Tween 80 addition) and diluted to 10^7^ cfu/mL for bacteria and 10^6^ cfu/mL for yeast and molds. In the next step, 1 mL of microorganism suspension was mixed with 1 mL of media: TSB (MERCK) for bacteria/MEB (MERCK) for microscopic fungi containing serial dilutions of the tested compounds. All samples were incubated at 37 °C for 24 h bacteria, 48 h yeast, and 28 °C for 48 h molds. As a growth control, a suspension of microorganisms in a medium without the biocides was used. The MICs were defined as the lowest concentration of the compounds in which there was no visible growth. All tests were repeated three times.

## 4. Conclusions

Gemini surfactants with a functionalized spacer by an ether group, i.e., 3-oxa-1,5-pentane-*bis*(*N*-alkyl-*N*,*N*-dimethylammonium bromides) were obtained in an innovative and ecological synthesis without using a solvent. The compounds show aggregation behavior characteristic of a homologous series of gemini surfactants. Surface activity increases (CMC decrease) with the elongation of hydrocarbon chain. The compounds 16-*O*-16 and 18-*O*-18 showed the highest aggregation activity (CMC ~ 0.03 mM). The standard Gibbs energy of micellization for all of the obtained GS take on negative signs, which means that the micellization process is spontaneous. Synthesized compounds also show characteristic dependence between antimicrobial activity and chain length, named cut-off effect. The most efficient microbiocide against all tested microorganisms is 12-*O*-12, which is comparable to the activity of compound 12-6-12. Compound 12-*O*-12 is a more than 30 times more efficient microbiocide than DTAB. Obtained GS are more active against Gram (+) bacteria than Gram (−) bacteria. These compounds constitute a new, interesting class of microbicides with a broad spectrum of biocidal activity.

## Figures and Tables

**Figure 1 molecules-26-05759-f001:**
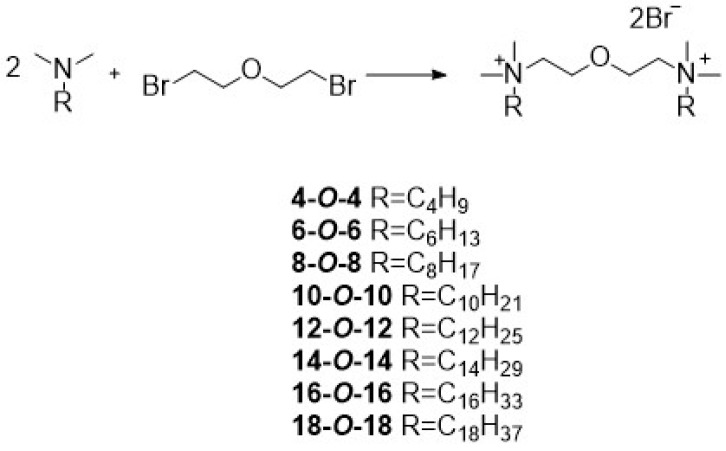
Synthesis of 3-oxa-1,5-pentane-*bis*(*N*-alkyl-*N*,*N*-dimethylammonium bromides).

**Figure 2 molecules-26-05759-f002:**
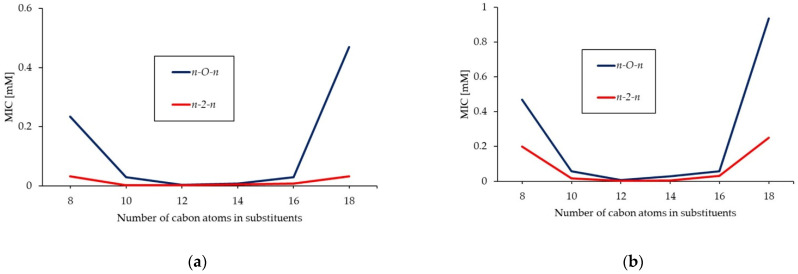
Quasi parabolic dependency of the antibacterial activity against *S. aureus* (**a**) and *E. coli* (**b**) on the hydrocarbon chain length for *n*-*O*-*n* (this research) and *n*-2-*n* [27].

**Figure 3 molecules-26-05759-f003:**
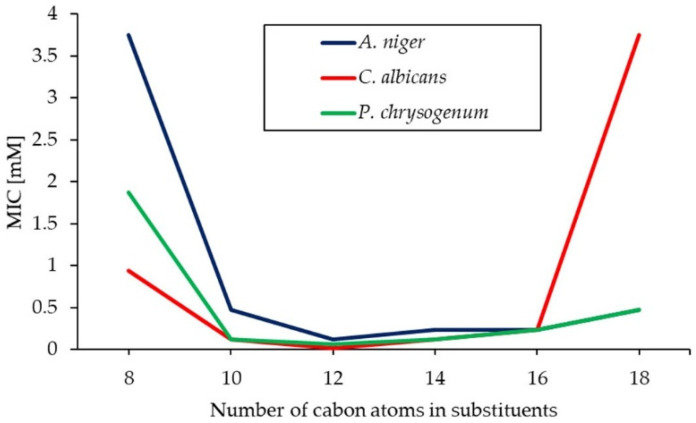
Quasi-parabolic dependency of the antifungal activity against *A. niger, C. albicans* and *P. chrysogenum* on the hydrocarbon chain length for *n*-*O-n*.

**Figure 4 molecules-26-05759-f004:**
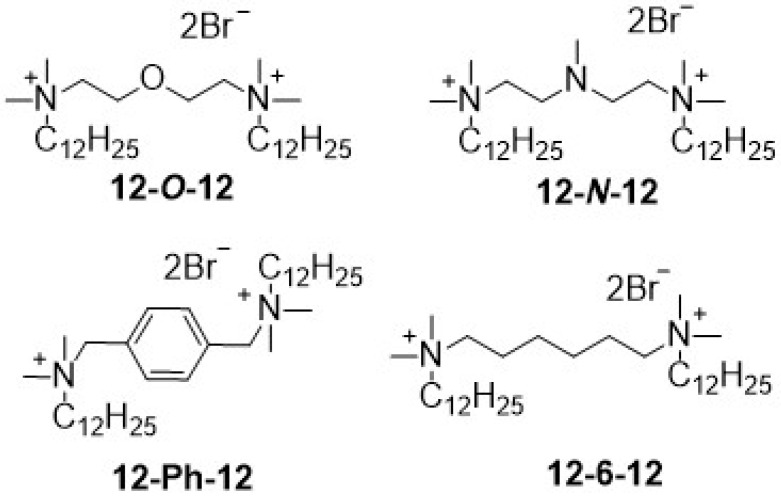
Chemical structure of GS with different spacer nature.

**Table 1 molecules-26-05759-t001:** Proton chemical shifts from ^1^H NMR spectra of gemini surfactants (*n*-*O*-*n*) molecules.

Gemini	-OCH_2_	-N^+^CH_2_CH_2_O-	-N^+^CH_2_-	-N^+^(CH_3_)_2_	-N^+^CH_2_CH_2_-	-CH_2_(CH_2_)_x_CH_3_	-CH_2_CH_3_
4-*O*-4	4.14 (4H)	3.83 (4H)	3.53 (4H)	3.29 (12H)	1.74 (4H)	1.44 (4H)	1.01 (6H)
6-*O*-6	4.18 (4H)	3.85 (4H)	3.51 (4H)	3.32 (12H)	1.75 (4H)	1.37 (12H)	0.90 (6H)
8-*O*-8	4.35 (4H)	4.04 (4H)	3.65 (4H)	3.46 (12H)	1.73 (4H)	1.47–1.19 (20H)	0.88 (6H)
10-*O*-10	4.36 (4H)	4.04 (4H)	3.65 (4H)	3.46 (12H)	1.73 (4H)	1.48–1.14 (28H)	0.88 (6H)
12-*O*-12	4.35 (4H)	4.04 (4H)	3.63 (4H)	3.46 (12H)	1.73 (4H)	1.45–1.17 (36H)	0.88 (6H)
14-*O*-14	4.36 (4H)	4.04 (4H)	3.65 (4H)	3.46 (12H)	1.73 (4H)	1.49–1.08 (44H)	0.88 (6H)
16-*O*-16	4.36 (4H)	4.05 (4H)	3.64 (4H)	3.46 (12H)	1.73 (4H)	1.47–1.10 (52H)	0.88 (6H)
18-*O*-18	4.16 (4H)	3.84 (4H)	3.49 (4H)	3.29 (12H)	1.72 (4H)	1.46–1.03 (60H)	0.88 (6H)

**Table 2 molecules-26-05759-t002:** Carbon chemical shifts from ^13^C NMR spectra of gemini surfactants (*n*-*O*-*n*) molecules.

Gemini	-N^+^CH_2_CH_2_O-	-OCH_2_-	-N^+^CH_2_-	-N^+^CH_3_	-N^+^CH_2_(CH_2_)_x_-	-CH_2_CH_3_	-CH_2_CH_3_
4-*O*-4	65.61	64.42	63.98	51.36	24.42	19.39	13.41
6-*O*-6	65.88	64.47	63.96	51.40	31.10, 25.70, 22.55	22.20	13.69
8*-O*-8	65.76	64.56	63.89	51.58	31.57, 29.18, 28.98, 26.23, 22.83	22.48	13.97
10-*O*-10	65.83	64.60	63.96	51.61	31.74, 29.38, 29.35, 29.25, 29.15, 26.28, 22.87	22.55	13.99
12-*O*-12	65.81	64.57	63.94	51.60	31.82, 29.53, 29.46, 29.38, 29.28, 29.26, 26.27, 22.87	22.60	14.05
14-*O*-14	65.87	64.61	64.00	51.64	31.85, 29.62, 29.60, 29.59, 29.56, 29.48, 29.40, 29.29, 26.31, 22.90	22.61	14.04
16-*O*-16	65.92	64.59	64.03	51.57	31.82, 29.61, 29.58, 29.56, 29.54, 29.45, 29.37, 29.25, 26.25, 22.83	22.58	14.00
18-*O*-18	65.65	64.56	63.76	51.37	31.84, 29.63, 29.61, 29.58, 29.52, 29.48, 29.40, 29.28, 26.31, 22.85	22.60	14.03

**Table 3 molecules-26-05759-t003:** Aggregation parameters for 3-oxa-1,5-pentane-*bis*(*N*-alkyl-*N*,*N*-dimethylammonium bromides).

Gemini Surfactants	CMC (mM)	β	ΔG^0^_mic_ (kJ/mol)
4-*O*-4	158.5 ± 3.5	0.31	−11.8
6-*O*-6	81.1 ± 2.0	0.34	−12.2
8-*O*-8	21.1 ± 0.9	0.34	−17.8
10-*O*-10	6.59 ± 0.16	0.51	−26.9
12-*O*-12	1.047 ± 0.006	0.71	−42.9
14-*O*-14	0.201 ± 0.002	0.58	−47.3
16-*O*-16	0.032 ± 0.002	0.44	−49.9
18-*O*-18	0.033 ± 0.001	0.58	−56.9

**Table 4 molecules-26-05759-t004:** MIC values (mM) for 3-oxa-1,5-pentane-*bis*(*N*-alkyl-*N*,*N*-dimethylammonium bromides).

Gemini Surfactants	MIC (mM)				
	*S. aureus*	*E. coli*	*C. albicans*	*A. niger*	*P. chrysogenum*
4-*O*-4	>3.75	>3.75	>3.75	>3.75	>3.75
6-*O*-6	1.875	1.875	>3.75	>3.75	>3.75
8-*O*-8	0.234	0.469	0.937	3.75	1.875
10-*O*-10	0.0293	0.058	0.117	0.469	0.117
12-*O*-12	0.0037	0.0073	0.0146	0.117	0.058
14-*O*-14	0.0073	0.0293	0.117	0.234	0.117
16-*O*-16	0.0293	0.058	0.234	0.234	0.234
18-*O*-18	0.469	0.937	>3.75	0.469	0.469

**Table 5 molecules-26-05759-t005:** MIC values (mM) of gemini surfactants with different spacer nature.

Gemini Surfactants	MIC (mM)				
	*S. aureus*	*E. coli*	*C. albicans*	*A. niger*	*P. chrysogenum*
12-*O*-12	0.0037 ^1^	0.0073 ^1^	0.0146 ^1^	0.117 ^1^	0.058 ^1^
12-*N*-12	0.003 ^2^	0.007 ^2^	0.023 ^2^	0.116 ^2^	-
12-Ph-12	0.0122 ^3^	0.0244 ^3^	0.0122 ^3^	0.0976 ^3^	0.0488 ^3^
12-6-12	0.0036 ^4^	-	0.015 ^5^	0.12 ^5^	0.06 ^5^

^1^ This study; ^2^ [101]; ^3^ [98]; ^4^ [22]; ^5^ [114].

## Data Availability

The data presented in this study are available in the article.

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
