# Peer review of "Antimicrobial Activity of Gemini Surfactants with Ether Group in the Spacer Part"

_molecules, 2021, doi:10.3390/molecules26195759_

Round 1

Reviewer 1 Report

It is a well-done work the aim of which was to obtain and test antimicrobial activity of gemini surfactants with ether group in the spacer part. The biocidal activity of this type of derivative has not been studied so far.The results are reliable, well described and very well discussed against other Gemini surfactants. The manuscript is well organized and written based on well choosen literature.

Author Response

Dear Reviewer,

Thank you very much for analyzing the work and for your positive assessment.

Reviewer 2 Report

In this work amonium quaternary gemini surfactants with an ether spacer were obtained, and the antimicrobial properties against bacteria, fungi and mold were investigated. The surfactants synthesized have a variety of hydrocarbon chain lengths which gives novelty and a scientific background that makes the research scientifically sound.

The paper is overall well written although has general English revision should be performed. Reference list is adequate and the abstract is well written. Now I will describe specific points

Line 29-30 Revise English pandemic of COVID-19 (Coronavrus Disease 2019)

Line 78 Revise English have been obtained in low-cost and green synthesis

Line 74-79 Explain better objectives and focus  of the research, rewrite this

Line 130- 147 I recommend adding the NMR and IR spectra in the supplementary file for better understanding

Line 178 Revise English

Line 198-202 English

Author Response

Dear Reviewer,

Thank you very much for reading and commenting on the manuscript.

We provide our explanations on the comments below:

  • line 29-30, it has been corrected
  • line 78, it has been corrected
  • line 74-79, it has been corrected
  • line 130-147, NMR and FTIR spectra were added in Supplementary Material
  • line 178, it has been corrected
  • line 198-202, it has been corrected

Reviewer 3 Report

The manuscript deals with the antibacterial activity of some gemini quaternary ammonium surfactant with an ether group in the sapcer. Overall, the manuscript is interesting, especially in highlighting the effect of 12-0-12 with other analogues (12-N-12; 12-6-1 and 12-Ph-12) from the literature.

Some comments are:

The chemical physical characterization of sufactant is poor and limited to conductimetric measurements for CMC determination. Surface tension analysis can be useful for the interpretation of antibacterial results.

No toxicity data are presented. Are MIC values toxic concentration for human cells or not? The toxicity strongly affects the employability of surfactants.

Some parts in the results section should be moved in the method section. Specifically, lines 82-84 and 94-104 for synthesis and lines 172-182 for conductimetric analysis and CMC determination.

The synthetic approch of the surfactants should be better addressed and highlighted in the manuscript.

 Greenolution is not a standard word. I suggest to write it in quotes as "greenolution" and define it at  the first appearance (green evolution).

Author Response

Dear Reviever,

Thank you very much for reading and commenting on the manuscript.

We provide our explanation on the comments below:

  • the interpretation of the results of  antimicrobial activity  is always based on the CMC which is most accurately determined by conductometric method
  • Gemini surfactant are typical microbiocides, not drugs; contact these preparations with people is very limited and poses no danger
  • line 82-84, 94-104 and172-182, have been improved
  • synthetic details were adeed in Supplementary Material

Round 2

Reviewer 3 Report

The manuscript is suitable for publication